# Is it really "panic buying"? Public perceptions and experiences of extra buying at the onset of the COVID-19 pandemic

Evangelos Ntontis[1,2]*, Sara Vestergren[3], Patricio Saavedra[4], Fergus Neville[5], Klara Jurstakova[2], Chris Cocking[6], Siugmin Lay[7], John Drury[8], Clifford Stott[3], Stephen Reicher[9], Vivian L. Vignoles[8]

1 School of Psychology and Counselling, The Open University, Milton Keynes, United Kingdom, 2 School of Psychology and Life Sciences, Canterbury Christ Church University, Canterbury, United Kingdom, 3 School of Psychology, Keele University, Keele, United Kingdom, 4 Instituto de Ciencias Sociales, Universidad de O'Higgins, Rancagua, Chile, 5 School of Management, University of St Andrews, St Andrews, United Kingdom, 6 School of Humanities and Applied Social Sciences, University of Brighton, Brighton, United Kingdom, 7 Centro de Medición Mide UC, Pontificia Universidad Católica de Chile, Santiago, Chile, 8 School of Psychology, University of Sussex, Brighton, United Kingdom, 9 School of Psychology and Neuroscience, University of St Andrews, St Andrews, United Kingdom

* evangelos.ntontis@open.ac.uk

**Data Availability Statement:** In this paper we report findings from a set of qualitative, semi-structured interviews. Due to ethical concerns and the limitations imposed to us by the ethics committee, we do not have permission to publicly

## Abstract

Shopping behaviour in response to extreme events is often characterized as "panic buying" which connotes irrationality and loss of control. However, "panic buying" has been criticized for attributing shopping behaviour to people's alleged psychological frailty while ignoring other psychological and structural factors that might be at play. We report a qualitative exploration of the experiences and understandings of shopping behaviour of members of the public at the onset of the COVID-19 pandemic. Through a thematic analysis of semi-structured interviews with 23 participants, we developed three themes. The first theme addresses people's understandings of "panic buying". When participants referred to "panic buying" they meant observed product shortages (rather than the underlying psychological processes that can lead to such behaviours), preparedness behaviours, or emotions such as fear and worry. The second theme focuses on the influence of the media and other people's behaviour in shaping subsequent shopping behaviours. The third theme addresses the meaningful motivations behind increased shopping, which participants described in terms of preparedness; some participants reported increased shopping behaviours as a response to other people stockpiling, to reduce their trips to supermarkets, or to prepare for product shortages and longer stays at home. Overall, despite frequently using the term 'panic', the irrationalist connotations of "panic buying" were largely absent from participants' accounts. Thus, "panic buying" is not a useful concept and should not be used as it constructs expected responses to threat as irrational or pathological. It can also facilitate such behaviours, creating a self-fulfilling prophecy.

share the full interview transcripts with anyone other than the authors of this paper. However, in line with the guidelines of PLOS for data availability, and particularly regarding qualitative data (https://journals.plos.org/plosone/s/data-availability), we have made publicly available all the anonymized related quotes and interview excerpts that we identified in the dataset during the coding phase of the analysis. All relevant interview excerpts can be found in the following link: https://osf.io/t43qx/.

**Funding:** The research presented here was supported by a QR seed grant by the School of Psychology and Life Sciences at Canterbury Christ Church University awarded to Evangelos Ntontis, and by a UKRI grant awarded to John Drury, Clifford Stott, Stephen Reicher, Fergus Neville and Evangelos Ntontis (ES/V005383/1). The funders had no role in study design, data collection and analysis, decision to publish, or preparation of the manuscript.

**Competing interests:** The authors declare that there are no potential conflicts of interest with respect to the research, authorship, and/or publication of this article.

## Introduction

At the onset of the COVID-19 pandemic in early 2020, newspapers in several countries published photos of empty supermarket shelves to illustrate shortages of food and other products [1–3]. Such observations were often characterized as instances of "panic buying" [3]. The underlying assumption was that perceptions of actual or expected shortages led people to behave irrationally and overreact, and that overreaction took the form of selfish behaviours and aggressive competition, eventually accentuating problems of supply for the public. However, only a very small percentage of shoppers bought very large amounts of products (e.g., 3% bought excessive amounts of pasta [4]). One major reason for product shortages was not due to irrationality but due to people's behavioural changes (e.g., many people increasing their purchases by a small amount in order to reduce trips to the supermarkets amidst the pandemic) that caused problems to modern fragile just-in-time supply chains [4, 5].

The aim of this paper is to explore the experiences of members of the public regarding product shortages during the onset of the COVID-19 pandemic as well as their understanding of the factors that can affect their own and other people's shopping behaviours. This study is important for various reasons. First, evidence shows that product shortages are predominantly related to structural factors [4, 5] rather than pathologies residing within people's psychology. Second, there is a rich history and implications of (mis)using pathologizing psychological discourses such as panic in relation to social behaviour [6, 7] but no significant attention has been paid to the notion of "panic buying", despite it being a concept frequently invoked by the media and in public discourse. Third, as we will show, previous studies have predominantly been quantitative with in-depth examinations of the phenomenology of experiencing and reacting to product shortages largely missing. The importance of our study lies in the argument that a better understanding of the motivations behind rapid spikes in demand can be crucial in building practical and sustainable solutions, in undermining potential for conflict, and in informing public policy and communication without reproducing myths about the driving forces of social behaviour.

### Factors associated with shopping behaviours in extreme events

At the onset of the COVID-19 pandemic, there were sharp increases in spending and reports of stockpiling of groceries, paper products [8] and medical supplies [9, 10]. There are two types of psychological explanations in relation to the factors that mobilized such behaviours; first, explanations that predominantly characterize the presence of empty shelves as instances of "panic buying" attributed to people's alleged psychological frailty. Second, explanations that consider socio-psychological factors such as risk perception and the influence of structural factors (e.g., demographic characteristics). Despite their significant differences and insightful findings as we will describe below, studies on shopping behaviours during extreme events including COVID-19 have largely avoided examining the phenomenology of such instances; that is, in-depth accounts of people's perceptions, experiences, and of the social contexts that shaped subsequent decision making are scarce.

**Pathologizing explanations of shopping behaviour.** Pathologizing psychological discourses featured heavily in lay accounts of shopping behaviours at the onset of the pandemic. Largely in absence of empirical data, media reports commonly attributed "panic buying" to "contagion of fear" [11], "amplified fear" [12], or "herd behaviour" [13]. "Herd behaviour" was understood as regression to a survival mode driven by primitive, animalistic instincts, and as obscuring rationality [13].

Empirical research on the psychological basis of shopping behaviours in extreme events remains relatively scarce, and at the onset of COVID-19, some academic accounts echoed

these pathologizing media discourses. Early scientific commentaries that addressed shopping behaviours at the onset of COVID-19 were mostly speculative and sometimes pathologizing. Some attributed shopping behaviours to people's attempts to reduce internal conflict and uncertainty or described it as a form of social influence driven by other people's behaviours and due to "contagion" of psychological responses [14]. Loxton and collaborators [15] partially attributed stockpiling to an emerging "herd mentality" and the disappearance of logic and individuality, as well as to the "Tragedy of the Commons", whereby people are overwhelmed by "panic" and selfish opportunism, acting against the collective interest. Similarly, Prentice, Quach and colleagues [16] conceptualized "panic buying" through the lens of classical, pathologizing theories of crowd psychology and behavioural contagion (e.g., [17]; for a critical overview see [6]). In other studies (e.g., [18, 19]), "panic buying" was used to describe fear, worrying, and uncertainty, without offering a precise definition of what "panic buying" means beyond these emotional responses.

Apart from the pathologizing elements that are common across many media and academic accounts as presented earlier, another shared similarity in that literature is a lack of personal accounts from people who were presented as "panic buying". Rather, journalists or academic authors of such accounts seem to be making *ad hoc* explanations of observed behaviour as external observers and in absence of empirical data on the subjective experiences of those involved.

**Psychological and structural factors as drivers of shopping behaviours in extreme events.** Rather than resorting to pathologizing concepts, some researchers have examined the role of non-pathologizing psychological and structural factors in driving shopping behaviours in extreme events. Some studies have drawn on game theory to argue that "panic buying" is based on people's understanding that their payoff is dependent on both their own and other people's behaviours [20]. However, Taylor [21] argues that, despite being potentially insightful, game theory is insufficient as an explanation for stockpiling as it cannot explain why only certain products become targets of stockpiling behaviours. The theory also cannot account for how contextual factors (e.g., the social environment or the information available to participants) can shape psychological reactions and behaviour [22]. For example, consumers' behaviour can be affected by observing other people's behaviour [23, 24]. Stockpiling may begin when other people's stockpiling behaviours cross a certain threshold, whereas behaviours perceived as under the threshold do not increase stockpiling demands. Consequently, seeing other people stockpiling or observing queues in supermarkets either in person or through media reports can generate uncertainty about one's own choices and beliefs, and facilitate stockpiling behaviours through a form of social learning [23].

The media can also play a significant role in creating and disseminating perceptions of high risk, eventually facilitating stockpiling behaviours [24]. Studying the effects of COVID-19 in New Zealand, Hall and collaborators [25] demonstrated that stockpiling occurred simultaneously with increasing media coverage and public awareness of the situation, lasting for less than a week. Similarly, Paek and collaborators [26] showed that exposure to emergency-related news was positively associated with the number of emergency-preparedness items owned. In consequence, it is often people who are less prepared for or who have experienced disasters in the past that tend to stockpile. Additionally, media reports often focus on isolated stockpiling behaviours, heightening market demands and further facilitating stockpiling [27]. Considering the aforementioned factors, Taylor [21] presents a model of how stockpiling escalates, arguing that as people visit stores to stock up for potential lockdowns, media reports exaggerate the significance of a minority of shoppers who might over-purchase. Such images in turn might become widespread and create perceptions of scarcity, eventually amplifying perceptions of scarcity and urgency among the majority. Thus, stockpiling is a snow-balling effect based on

the fear created by perceptions of scarcity and through observing other people's behaviour. Similarly, Sterman and Dogan [28] argue that stockpiling may be triggered by situational factors such as stressors created by perceptions of scarcity or poor supply delivery rather than idiosyncratic attributes residing within the individuals. These findings are in line with reviews of the literature showing that perceptions of scarcity and threat, as well as uncertainty due to fear of the unknown, can affect stockpiling behaviours [24].

Governmental policies, trust in government, as well as demographic and household conditions can affect shopping behaviours in extreme events [24, 29, 30]. An analysis of secondary data in Australia showed that shopping behaviours escalated in mid-March 2020 when more strict measures were introduced such as extended lockdowns and social distancing [31], whereas purchases remained at normal levels when smaller-scale measures were in place during the previous months [31]. Similarly, stocking up on supplies during extreme events can be driven by perceptions of restrictions in movement and the availability of supplies as well as expected price hikes [32, also see 33]. Bentall and colleagues [34] present a model that considers several of the aforementioned factors, showing that stockpiling in the UK and the Republic of Ireland was positively predicted by income, having children at home, psychological distress, threat sensitivity, and mistrust of others.

In contrast to pathologizing accounts that we briefly discussed in the previous section, the studies discussed above considered the psychological and structural factors that can affect shopping behaviour in extreme events. For instance, some of the factors considered were related to social trust and trust in the government, demographic characteristics, other people's behaviour, as well as the role of the media in creating perceptions of scarcity and potentially facilitating stockpiling. However, the vast majority of research that addresses shopping behaviours in extreme events, even if not pathologizing and irrationalist in nature, is mostly quantitative and focuses on modelling the predictors of such behaviours. What we have not identified are phenomenological studies that provide in-depth accounts of people's subjective experiences and perceptions regarding shopping behaviours in extreme events. Such studies are important as they can provide insights on the social context within which people's decision-making processes occurred, give voice to those involved to explain the rationale behind their actions, and can help identify further factors to be subsequently tested through quantitative methods.

## The present study

Our aim in this paper is to examine the perceptions and experiences of members of the public in relation to product shortages at the onset of the COVID-19 pandemic. More specifically, we are interested in accounts of members of the public not only to examine the range of experiences, understandings, and reactions to product shortages but also because they allow us to explore the narratives employed to account for personal and other people's behaviours. This focus has been largely absent from previous studies due to their emphasis on quantification and prediction of various associated variables rather than on the lived experience of the phenomenon.

## Method

### Participants, recruitment, and interview questions

To explore subjective experiences, we carried out semi-structured interviews with 23 members of the public. Our only inclusion criteria were that participants were over 18 years old and resided in the United Kingdom at the onset and through the early course of the pandemic. Residing in the UK during the pandemic was important as it ensured that participants were

subject to similar news reports and governmental announcements and had broadly similar shopping experiences and exposure to other people's behaviours. We used opportunity sampling and participants were recruited through personal contacts, snowballing, and social media (Facebook, Twitter) on the basis of their willingness to participate in our study. Thus, the sample is not representative of the UK population. Following the initial interviews, we asked participants to introduce us to further contacts that might be willing to participate. Nineteen participants were interviewed individually, and 4 were interviewed as couples based on their preferences. Eighteen participants were females and 5 were males, and all resided in England (Southern, Midlands, Northern) and Scotland. Participants' ages ranged between 23 and 76 years old ($M$ = 36.4 years, $SD$ = 15.1) [exact ages of two participants approximately in their 40s and 50s are not available]. There was variation in participants' employment status; full time or part-time employees, self-employed, retired, unemployed, or doctoral students. The total duration of the interviews was 567 minutes ($M$ = 27.01 minutes, $SD$ = 9.74). Due to restrictions in collecting face-to-face data in May 2020, interviews were carried out online via MS Teams or Zoom. Only one interview was carried out at an outdoor space with participants wearing masks and keeping appropriate physical distancing. Ethical approval for the study was obtained from the Research Ethics Committee of Canterbury Christ Church University (ETH1920-0200). Before the interviews commenced, participants had to provide the interviewers with their consent. Since the interviews were carried out online or while observing social distancing measures, a consent form was sent to participants in advance electronically together with a participant information sheet which described the aims of the project. Participants then had to confirm verbally that they read and understood the information sheet, that their participation was voluntary and that they could withdraw at any point without the need to provide a reason, that they agreed to the interview being audio recorded, that their information would be kept strictly confidential in line with the lead author's University Research Privacy Notice, and that they agreed to take part in the project overall.

Interviews were conducted by the first, second, fourth, fifth, and sixth authors. The first author drafted an interview schedule which was revised carefully by the whole team before data collection started. Interview questions included whether and how the pandemic has affected participants' lives (Has the pandemic affected you in any way?), their reactions and feelings (How did you react when you first heard about the pandemic in your country/local area?), ways of preparing (Did you prepare/are you preparing in any way to protect against the outbreak?), concerns about shortages of supplies and shopping habits (Do/did you worry that there might be a shortage of food or other goods? Why yes/no?), reactions towards government announcements (Did the official announcements have an effect on your preparation?), and perceptions of media and of other people's perceptions and behaviours (Did social media have any effects on your preparation? Did other people's behaviour have any effect on your preparations?). The complete list of interview questions can be found in the Supporting Information provided. Considering that "panic buying" is not a neutral concept and is loaded with assumptions regarding the fragility of human psychology, we intentionally did not include it in our interview questions and only referred to it when participants had already suggested it themselves in order to further explore the meanings attributed to it. The term was only used in one interview question, in which we asked participants whether they believed the government statements that there was enough food and there was no need to "panic buy". This question was developed in response to a statement made by the UK's Prime Minister Boris Johnson in March 2020 that "people should have no need to stockpile or to panic buy" [35]. All interviews were audio-recorded and then transcribed verbatim.

### Analytic procedure

Our analysis was exploratory and aimed to understand how participants understood "panic buying" as well as to delineate the factors that participants perceived to shape their own as well as other people's shopping behaviours. We performed a reflexive thematic analysis in our dataset [36]. After the interviews were transcribed verbatim by a transcription company, we conducted multiple readings of the data corpus while taking initial notes on issues related to our research question. In subsequent readings of the data, we created codes (analytic units; [37]) that helped us capture various facets of our observations. Such codes included various emotions reported by participants, personal and other people's reactions to the pandemic, or different meanings of "panic buying". We then compared and critically reflected upon those codes and their interrelationships which finally led to the development and naming of the final themes presented in the following section. Our analytical claims are supported by quotes from multiple participants. The symbol [. . .] denotes text which has been removed to ease readability. Participants appear by their respective number (e.g., P1 stands for 'Participant 1).

## Findings

We constructed three themes based on the transcripts: The first theme concerns participants' definitions of "panic buying". The second theme refers to participants' perceptions of factors contributing to certain shopping behaviours and mainly revolves around the effects of social media and other people's behaviour. Finally, the third theme addresses the element of preparedness and the various reasons that led participants to stockpile. A summary of the themes is presented in Table 1 below.

### What do people mean by "panic"? Product shortages, preparedness, fear, and uncertainty

The notion of "panic" appears frequently in media and public discourse and was often employed by our participants to explain or define the observed phenomena of increased shopping during the early stages of the COVID-19 pandemic. As this theme demonstrates, what participants understood when using the concept of "panic" was not what is considered as irrationality and loss of control but rather observations of a lack of products or expected feelings of fear or uncertainty. The extract below illustrates how "panic" was equated to a shortage of products:

Extract 1

*I*: *Yeah. When you say people panicking, what do you mean, more or less*?

*P2*: *Like the bulk buying of toilet roll and taking all the long-life stock off the shelfs*.

*P1*: *Yeah, like all the canned food was gone [. . .] And oil and flour and pasta and stuff [. . .] And it was quite hard. I think, initially, like the first two weeks to find that in a lot of shops*.

**Table 1.  Table of themes identified in the dataset.**

| Theme 1 | *What do people mean by "panic"? Product shortages, preparedness, fear, and uncertainty* |
| --- | --- |
| Theme 2 | *Shopping behaviours as influenced by media reports and other people's behaviour* |
| Theme 3 | *The meaningful motivations behind increased purchases: Reducing trips to supermarkets, preparing for product shortages and longer stays at home, or guarding against "those who panic"* |

Due to its diffusion in both lay and scientific discourses, the notion of "panic" was readily available for participants to employ as an explanation for shopping behaviours, compared to alternative discourses that were less accessible. For example, participants often used "panic" and "panic buying" to describe instances where there were perceptions or expectations of no or little resources available for prolonged periods of time rather than a psychological state of irrationality as "panic" connotes. In other instances, participants used "panic" to describe feelings of fear or worry:

Extract 2

*I*: *And why do you think that is*? *[people buying more supplies than usual]*

*P20*: *Panic. I think people were fearful that they weren't going to be able to feed themselves. And I think with some people there was, you know, "We have to get it first." Yes, there were a lot of people that seemed to, sort of, go into selfish mode. [. . .] I would argue that people probably become more heavily influenced by what's going on around them when they go into panic mode.*

Extract 2 demonstrates how 'panic' in participants' accounts is underpinned by fear that stems from a need to survive and perceptions of how other people acted. The use of "panic" to describe an underlying emotion was very common throughout our interviews. It is important to emphasise that when our participants used "panic" to explain or define the phenomena, it was mostly in reference to others' behaviour (cf. [38]). When prompted to further discuss their explanations, participants related the fear or worry to various sources; the uncertainty of the situation, need for survival or risk management. Hence, when explaining the psychology behind "panic" our participants simply referred to expected psychological reactions in a crisis (e.g., fear) rather than a loss of control.

In other cases, such as in the following two extracts, participants associated "panic" with peoples' fear and worry due to uncertainty regarding the short- and long-term impact of the pandemic. "Panic buying" was often equated with the uncertainty of access to products and one's economic situation, pointing to an underlying rationale for the behaviour of increased shopping. The increased shopping could thereby be a way for people to take control in an uncertain situation or at least to prepare and remain safe when confronting the unknown:

Extract 3

*P3*: *There was quite a lot of panic, I would have said.*

*I*: *Okay. In what way*? *What do you mean by panic*?

*P3*: *Well, people didn't know whether they would be able to get what they needed and they were not sure if they were going to have, they wouldn't, they didn't know what kind of impact it was going to have on their economic security.*

And

Extract 4

*I: Yeah. So you mentioned panic buying, why do you think people do- first of all, what do you mean by panic buying, how do you define it more or less*?

*P6: I think it comes from people being scared that there won't be any resources anymore so they start yeah, having to stock everything at home so they can, if yeah, I don't know, the end*

*of the world really happened that they will have food for several weeks to survive, so I guess it comes from being afraid of the unknown*

Overall, this theme captured the different meanings that participants attributed to "panic buying". As it becomes evident from the data, illustrated through the quotes above, "panic" was not used to connote irrationality and loss of control–rather, our participants used the concept to refer to either observations of a lack of products or to expected feelings of fear and worry due to uncertainty and associated acts of preparedness. What becomes apparent is that "panic" is not a useful concept as it does not meaningfully describe observed behaviours or their underlying psychology.

The next theme is associated with participants' perceived factors that can mobilize shopping behaviours; the main reported factors were reports from the media as well as observing other people's behaviour.

## Shopping behaviours as influenced by media reports and other people's behaviour

Participants reflected extensively on the reasons that might motivate shopping behaviours. Two elements that featured prominently were the media and observing other people's behaviour:

Extract 5

*I*: *So, do you think, in terms of preparations, do you think that people generally bought more supplies than usual*?

*P3*: *Yes.*

*I*: *Yeah. So why do you think they did that*?

*P3*: *Because they didn't know when they were going to be able to get some more. And the more, the more you have media reports saying we're running out of something, the more people think, well, I'll get two if I see them.*

Our participants agreed that people engaged in increased purchases. However, the explanation offered was not associated with irrationality or loss of control as traditional "panic" discourses suggest. Rather, they argued that initially, people bought more products because they were uncertain regarding future availability. Uncertainty and subsequent increased purchases were compounded by media reports showcasing product shortages. Such media reports were represented as motivating people to slightly increase their purchases. In other cases, illustrated for example by extract 6 below, media reports were represented as fostering perceptions regarding other people's behaviours:

Extract 6

*P2*: *I think a lot of panic buy, was fuelled by the media, of the panic buying.*

*I*: *Okay.*

*P2*: *So because you see it on the news and you'd go, oh, oh a lot people went, "oh, well I have to go out and buy these things".*

*P1*: *Yes.*

*P2*: *"Because everyone else is" [. . .] "And I need to get there first so that I can have some, instead of arriving there last and not getting any".*

Participants also argued that the media can motivate shopping behaviours through stories of how other people are behaving. In addition to media reports, our participants emphasised other people's behaviours, and more specifically metaperceptions of other people's ways of navigating through the pandemic, also featured more generally as a means of explaining instances of shopping behaviours:

Extract 7

*I*: *Yes. So, do you think generally that people bought more supplies than usual?*

*P8*: *Of some things, yes, definitely, because otherwise the shelves wouldn't be empty, yes.*

*I*: *Yes. Why do you think they do that?*

*P8*: *Yes, I know. It's difficult to figure that one out because I think because they're worried about what other people are going to do. They might not do it, but they just think that other people will buy all of it and so they do it. Then it becomes true, kind of, but at the beginning it's because everyone thinks, "Oh, my God, everyone's going to buy eggs." Then they buy eggs, but they don't need the eggs and they wouldn't do it themselves unless other people. . . Unless they thought other people would do it.*

Participants argued that despite initially being unwilling to shop for extra products, their perception that others will engage in increased purchases led them to buy additional products that they might not need.

Overall, in this theme, we highlighted the importance of media reports and other people's behaviours in guiding shopping behaviours and potentially causing problems in supply chains. What is striking once again is that participants did not use repertoires of irrationality to account for how other people behave. Rather, they mainly resorted to identifying either sources of influence (e.g., media reports) or people's speculations about other people's motives in driving particular behaviours, both of which point to meaningful social action. We will now turn to our third and last theme which further explores the elements of preparedness and risk perception. These concepts, which featured heavily in participants' accounts, were largely devoid of irrationalist narratives both in terms of personal as well as other people's behaviour.

## The meaningful motivations behind increased purchases: Reducing trips to supermarkets, preparing for product shortages and longer stays at home, or guarding against those who 'panic'

Preparedness featured heavily in participants' accounts of their experiences at the early stages of the pandemic. As this theme highlights, participants reported shopping behaviours through explanations of fear for potential product shortages, reduction in their trips to the supermarkets to reduce their exposure to the virus, preparation for longer stays at home (because of the lockdown or due to potential illness), or fear of other people 'panicking'.

Our participants used these rationalisations for understanding their own and others' shopping behaviours. One risk that participants argued that they were trying to protect themselves against was that of potential product shortages. This is illustrated by accounts from P1 and P2, as well as in the statement of P10 below:

Extract 8

*I: In terms of shopping behaviours, why did you start shopping for longer time, as you said like a, two weeks in advance.*

*P1: I think there was*

*P2: I was, so it was very much just in case we weren't able to- be able to access*

*I: Yeah.*

*P2: shops like for example with, we've only got small shops nearby us.*

*I: Yeah.*

*P2: And it could've been closed. There was also like the mad panic about certain things not being in supply and not coming into the country.*

*I: Okay.*

*P2: But it was just like a case of, if we act now, it makes our lives easier in the future.*

*P1: Yeah.*

*P2: While still being done in a sensible controlled manner rather than panic shopping.*

And

Extract 9

*P10: if somebody has a family and has to keep feeding their children, then they can't take the risk that there won't be enough food in a few months. So they have lower risk tolerance.*

Participants stated that their shopping patterns changed, and they started making larger and less frequent shops. The main reason provided was one of perceived inability to access shops as well as warnings about potential items not being in stock. However, participants were careful in framing increased shopping as a means of rational future preparedness rather than as irrational behaviour. Similarly, when referring to other people's increased purchases, participants attributed increased shopping to lower risk tolerance that some people might have due to various life conditions. For example, having to feed one's family can result in increased purchases under conditions of a perceived potential shortage of products.

The second reason that was commonly used in our participants' responses for increasing their shopping was the inclination to reduce trips to the supermarkets as well as because of actual changes in their routines (such as prolonged stays at home) due to restrictions in movement placed by governments:

Extract 10

*I: Yes. So, in general, would you say that you bought more than you would usually buy?*

*P11: Yes. Also, in the sense that, instead of going to the shop maybe two or three times a week in smaller amounts, I would just go once a week. So, obviously, I would buy more but overall, the amount would be the same. Also, because I was at home and cooking the whole time rather than going out, sometimes. So, I bought more but the amount of food I've consumed stayed constant.*

Preparedness manifested in participants' accounts concerning their wish to reduce their trips to the supermarkets. In some cases, more frequent trips to supermarkets with smaller

purchases were replaced by less frequent trips and larger purchases. On other occasions, there were reports of increased purchases compared to before the pandemic.

Extract 11

*I*: *But overall would you say that you bought more than you would usually buy*?

*P14*: *Yes. Maybe 5% and 10% more.*

*I*: *What do you think might be the main reason for that*?

*P14*: *I think the stuff I was saying before about wanting to extend how long it was between shops was the main factor. I was very aware that I did not want to run into the supermarket, take all of the tinned tomatoes, and leave none for anyone else. I wanted to feel that I had found a balance between feeling we had a bit of a buffer and still leaving enough for other people to have a buffer.*

The main reason, as illustrated by P14 above, was similarly associated with reducing one's trips to the supermarkets. However, the latter was compounded by wanting to provide oneself "a buffer" through means of some additional stocks at home while allowing other consumers to also increase their purchases slightly. Overall, reducing the number of trips to the supermarkets was a main reason that participants reported buying more items, either by actually slightly increasing their purchases or by making larger and less frequent shops while not changing the amounts. A third motivation for participants' preparedness was associated with their perceived potential need to stay at home for prolonged periods:

Extract 12

*I*: *Did you prepare in any way to protect yourself against the outbreak, or are you preparing now*?

*P10*: *Not really. I was not hoarding stuff or anything like this. I did some estimates, quick in my head, how much food will I have to buy in order to not have to leave my house for a month or two months*

When asked whether they prepared in any way against the outbreak participants, as illustrated by P10 above, denied hoarding and contested any notions of irrationality in their behaviour. Rather, they argued that their response was based on a meaningful calculation whereby they bought supplies that would last for approximately one or two months in case they could not leave the house for prolonged periods. Participants reported deciding to shop large quantities of products as a means of preparedness for staying at home for extended periods. Participants also reported considering either actual or potential health issues that might limit their future ability to leave their houses for prolonged periods:

Extract 13

*I*: *Okay. But [inaudible] beginning of it, did you start buying more things and usually in the beginning*? *At least for some goods as you said*?

*P3*: *I tried to make sure that we would have enough not to leave the house for a bit. I didn't know how long we would be stuck inside. Also, I have asthma, so the NHS was originally advising anybody who had a list of health conditions to, require you to have a flu jab. They said if you're on this list of conditions, which include asthma, then you should treat this as very serious and protect yourself. So, I've been [inaudible] cautious anyway.*

And

Extract 14

*P15*: *Well I, I didn't kind of see it as panicking and I don't think people were. So I know my neighbours, other people I see around my building, they were also trying to stock up and I didn't see that as panicking, they were just thinking, "If we're not allowed to go out and we're all being quarantined if I feel sick then this is what we have to do."*

Illustrated by the extracts above, participants expressed concerns with prolonged isolation at home due to illness. For example, they bought more items due to her medical condition which could potentially put them at risk of further complications if they caught COVID. Thus, participants decided to buy more items and reduce their trips to the supermarkets to reduce the risk of contracting the virus. Participants also reported how their decision-making and behaviour was also a reaction to the NHS guidelines for reducing risk and staying protected. Finally, preparedness was used by participants as a reason for being adequately stocked. However, they refrained from framing their actions as panic. Rather, they perceived both themselves as well as others stocking up in case they could not leave their houses due to quarantine measures.

From the examples presented here, it becomes clear that participants largely perceived themselves as well as other people as rational social actors who took preparedness measures in the face of uncertainty. However, there were some cases in which preparedness (and by extent some instances of shopping behaviours) was driven by people's perceptions of how other people would act:

Extract 15

*I*: *Okay. Did you believe any statements that you heard about there being enough food or not enough food and that there was no need to panic*?

*P19*: *Yes. I believed that there was no need to panic. My concerns weren't whether there was enough food and stock. It was whether we would be able to access it once people had started stockpiling.*

And

Extract 16

*P6*: *I didn't, so at some point I did but the thing is when- so I remember still going to the supermarket on a Saturday, a week or two before the lockdown, and everything seemed pretty normal although I noticed that the toilet paper aisle was empty, so I just did my shop that I usually do and then at around lunchtime I saw a message from our Facebook from the same supermarket that I've just been in and everything shelf was empty, everything, and that was, and that changed my mentality a bit in that way that I just got worried that just because everyone else is buying stuff I may not get a hold of it, so then I reached the point where I did my buy two instead of one, yeah, I don't know, tomatoes or pasta or whatever, just so I am on the safe side, but it was more- I would say this is more the second wave response because this was a response to other people panic buying.*

As illustrated above, participants reported buying additional products as a means of preparedness against potential shortages. However, from their accounts, their reactions were

largely driven by other people's behaviour. Participants' concerns were guided not by potential product shortages but due to other people buying extra products, which could potentially render them unable to purchase the products that they needed. Similarly, participants reported that they doubled how many products they bought not as a response to the pandemic itself but due to people's actual behaviours. The increase was driven by comparisons of the pre- and post-lockdown observations of shortages of some products which were amplified by messages in social media. Thus, some participants did increase their purchases as they were concerned about how other people would react to the pandemic.

## Discussion

The concept of "panic buying" has diffused into both scientific and lay discourses as a means of explaining shopping behaviours in extreme events and was used for example when empty shelves were observed at the onset of the COVID-19 pandemic (e.g., [3]). Some social scientists used a similar irrationalist discourse and attributed shopping behaviours to an emerging herd mentality [15] or uncritical contagion [16] or equated psychological responses which are expected in extreme events such as fear and worrying to irrationality and "panic" [19]. However, in the wider social scientific literature related to disasters, the notion of "panic" has been discredited as it is often used in an *ad hoc* manner and is not able to predict or explain observed behaviours of interest [7, 39–41]. Similarly, reports at the onset of the COVID-19 pandemic showed that only a very small percentage of shoppers bought very large amounts of products (e.g., 3% bought excessive amounts of pasta) whereas most people slightly increased their shopping which caused problems in supply chains [4].

In this paper, we were interested in people's experiences and understandings of shopping behaviours in extreme events. We wanted to explore both participants' experiences and perceptions of their own and other people's behaviour in the early stages of the COVID-19 pandemic, as well as how the concept of "panic buying" might be employed by participants themselves as an explanatory concept for observed behaviours. The thematic analysis based on semi-structured interviews provided us with in-depth insights on how people define "panic buying" as well as on the factors that affected their behaviour or were thought to have shaped other people's behaviour. Our first theme showed that "panic buying" was often utilized by participants as an explanatory concept for buying additional products. However, upon closer inspection, we noticed that the concept was either used to provide an *ad hoc* description of observed empty shelves or was employed to describe preparedness activities or emotions that normally emerge in emergencies such as fear and worrying, adding no further useful information. In our second theme, we showed that our participants perceived stockpiling as being affected by media reports which can shape expectations about how other people will behave, or due to the perceived or observed behaviour of other people. Media reports that depicted a lack of products represented other people's behaviour as essentially one of competition for access to a limited number of products, creating a sense of urgency to prevent one from being left out of essential products in times of crisis. When used to guide one's behaviour, this form of reasoning whereby one observes or speculates about how others will think and behave can eventually lead to a vicious circle, subsequently leading to supply chain issues. Our third theme demonstrated that the element of preparedness was crucial in shaping participants' responses or their perceptions of how and why other people were potentially stockpiling. Reducing one's trips to supermarkets, preparing for product shortages or for prolonged stays at home, or guarding against "those who panic" were perceived as the motivators for increasing one's purchases. It is worth noting that any increases reported by our participants were small and unlike the hoarding behaviours often reported in the media.

Considering our findings as well as reports that only a tiny minority of shoppers bought very large amounts of products [4], we argue that what might count as "excessive" can also lie in the eye of the beholder and can be affected by personal circumstances and perceptions shaped by the surrounding social context; for example, the amount of products that a parent of a family of five might purchase when preparing for a two-week-long lockdown can be perceived as a necessary act of preparedness by the parent but as "excessive" or irrational by an outsider such as a journalist reporting consumer behaviour at the onset of an extreme event. Our findings are in line with previous research on purchasing behaviours in extreme events. We highlighted the role of perceptions of scarcity and other people's behaviours [23, 28], distress and mistrust of others [34], and the media [26] in driving purchasing behaviours. Also, as expected, we found that participants often used "panic buying" to account for various observations and behaviours, but the concept was of limited use. Moreover, "panic buying" and its irrationalist connotations were potentially more dangerous when they were uncritically adopted in scientific reports [15, 16] as they distorted meaningful social action by pathologizing it.

In line with other researchers [7, 39–41], we argue that the notion of "panic" and "panic buying" should not be used to describe purchasing behaviours in extreme events for four reasons: *First*, it pathologises meaningful, adaptive social action by portraying it as irrational. *Second*, it does not offer any meaningful insights as it can only provide inaccurate, *ad hoc* explanations of observed behaviours. *Third*, it reproduces the myth of "panic" in both lay and scientific discourses and can set up problematic generalized representations of human behaviour, becoming dangerous when used to inform both media reporting as well as public policy. *Fourth*, attributing product shortages to human behaviour and more specifically to "panic" is problematic, since observed shortages are short-lived and are mostly caused by problems in the supply chains, by bulk shopping, or by small increases in the purchases of a small proportion of shoppers [4, 5].

## Limitations and future research

Our findings should be interpreted in light of the limitations of our study. The main issue concerns participants' reflections on their behaviour. Buying extra products is usually seen as antisocial and anti-normative behaviour, so even if participants did engage in hoarding, it is possible that they would not report it to us. However, many participants admitted sometimes buying more products than usual, so distortion of information is quite unlikely. Second, the interviews were collected in June and July 2020, at least three months since the first lockdown was imposed in the UK. The passing of time and the return of 'normality' might have changed participants' perceptions compared to the very early days of the lockdown, subsequently affecting their responses to the survey. Third, our sample was relatively small (but nevertheless typical for semi-structured, qualitative interview studies), young, and not representative in terms of age, socioeconomic status, or geographical location. These factors might have affected participants' risk perceptions and risk tolerance, affecting the findings. Based on the aforementioned limitations, future research should collect data at the time that stockpiling behaviours are observed and from larger and representative samples. Also, there is scope for research that will use both qualitative and quantitative survey findings to address the issue of stockpiling in experimental settings.

## Conclusion

Despite often using the term "panic buying", what our participants often meant were feelings of fear and uncertainty, the element of preparedness, or observed product shortages. A

potential reason why the discourse of "panic" was often used was not due to its insights but rather because of its immediate availability through its diffusion in popular culture [7, 38]. Moreover, the main reasons why participants reported increasing their purchases were not a loss of rationality but rather the influence of the media and of other people's behaviours which led them to take up preparedness action. The latter involved reducing trips to supermarkets, preparing for product shortages and longer stays at home, as well as protecting themselves from people who were perceived or represented as "panic buying". What becomes obvious is that human behaviour is not irrational but is controlled, adaptive, and imbued with (social) meaning. Thus, a cultural change is needed whereby social actors (e.g., the media, scientists, politicians) abstain from using language that pathologizes human behaviour and facilitates unnecessary competition.

## Supporting information

**S1 File. List of interview questions.**
(DOCX)

## Author Contributions

**Conceptualization:** Evangelos Ntontis, Sara Vestergren, Patricio Saavedra, Chris Cocking, Siugmin Lay, John Drury, Clifford Stott, Stephen Reicher, Vivian L. Vignoles.

**Data curation:** Evangelos Ntontis, Sara Vestergren, Fergus Neville, Klara Jurstakova, Chris Cocking, Siugmin Lay.

**Formal analysis:** Evangelos Ntontis, Patricio Saavedra, Fergus Neville, Chris Cocking, John Drury, Clifford Stott.

**Funding acquisition:** Evangelos Ntontis, Fergus Neville, John Drury, Clifford Stott, Stephen Reicher.

**Investigation:** Evangelos Ntontis, Sara Vestergren, Klara Jurstakova, John Drury.

**Methodology:** Evangelos Ntontis, Sara Vestergren, Patricio Saavedra, Fergus Neville, Siugmin Lay.

**Project administration:** Evangelos Ntontis, Patricio Saavedra.

**Validation:** Evangelos Ntontis, Sara Vestergren, Patricio Saavedra, Klara Jurstakova, Chris Cocking, Siugmin Lay, John Drury, Clifford Stott, Vivian L. Vignoles.

**Writing – original draft:** Evangelos Ntontis.

**Writing – review & editing:** Evangelos Ntontis, Sara Vestergren, Patricio Saavedra, Fergus Neville, Klara Jurstakova, Chris Cocking, Siugmin Lay, John Drury, Clifford Stott, Stephen Reicher, Vivian L. Vignoles.

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
