## [Decision Letter · Decision Letter 0]

2 Jan 2022

PONE-D-21-36415Is it really “panic buying”? Public perceptions and experiences of extra buying at the onset of the COVID-19 pandemic

Dear Dr. Ntontis,

Thank you for submitting your manuscript to PLOS ONE. After careful consideration, we feel that it has merit but does not fully meet PLOS ONE’s publication criteria as it currently stands. Therefore, we invite you to submit a revised version of the manuscript that addresses the points raised during the review process.

We look forward to receiving your revised manuscript.

Kind regards,

Mumtaz Alam, PhD

Academic Editor

PLOS ONE

Journal Requirements:

a) Did participants provide their written or verbal informed consent to participate in this study?

“The research presented here was supported by a QR seed grant by the School of Psychology and Life Sciences at Canterbury Christ Church University awarded to Evangelos Ntontis, and by a UKRI grant awarded to John Drury, Clifford Stott, Stephen Reicher, Fergus Neville and Evangelos Ntontis (ES/V005383/1).”

“The research presented here was supported by a QR seed grant by the School of Psychology and Life Sciences at Canterbury Christ Church University awarded to Evangelos Ntontis, and by a UKRI grant awarded to John Drury, Clifford Stott, Stephen Reicher, Fergus Neville and Evangelos Ntontis (ES/V005383/1).”

“The research presented here was supported by a QR seed grant by the School of Psychology and Life Sciences at Canterbury Christ Church University awarded to Evangelos Ntontis, and by a UKRI grant awarded to John Drury, Clifford Stott, Stephen Reicher, Fergus Neville and Evangelos Ntontis (ES/V005383/1).”

Reviewers' comments:

Reviewer's Responses to Questions

**Comments to the Author**

1. Is the manuscript technically sound, and do the data support the conclusions?

Reviewer #1: Yes

Reviewer #2: Yes

2. Has the statistical analysis been performed appropriately and rigorously? 

Reviewer #1: Yes

Reviewer #2: Yes

3. Have the authors made all data underlying the findings in their manuscript fully available?

Reviewer #1: Yes

Reviewer #2: Yes

4. Is the manuscript presented in an intelligible fashion and written in standard English?

Reviewer #1: Yes

Reviewer #2: Yes

5. Review Comments to the Author

Reviewer #1: The study is quite interesting and also it is well-written article. I have thoroughly gone through the whole article and found it to be quite interesting paper with good policy implication. Authors have worked alot on all the sections of the paper and thus, I strongly support this article for publication without any further changes.

Reviewer #2: I would first like to extend appreciation for your work. Interesting set of thematic analyses. Please find below minor comments/questions:

1. Consent statements have not been included in your manuscript. While it is good that you've included the ethical statement, it would also be important to include the consent statement as well.

2. Can the authors clearly state their exclusion criteria? (Before I read your "limitations" segment, I assumed that you deliberately focused on a specific demographic)

3. I believe you must address the sample size as a limitation maybe, to note that it is in fact to small to be used for generalization of the topic under discussion.

6. PLOS authors have the option to publish the peer review history of their article (what does this mean?). If published, this will include your full peer review and any attached files.

Reviewer #1: No

Reviewer #2: No

---

## [Author Response · Author response to Decision Letter 0]

7 Jan 2022

Our responses to the comments raised by the reviewers and the editor have been addressed in the response letter. However, I also copy/paste them here:

We would like to thank the editor and the reviewers for their positive comments on our paper as well as for the opportunity to resubmit. In this letter, we address all comments raised. Our responses appear in red below the comments. 

Kind regards,

The authors

We have added information in the authors’ affiliations. Also, we have removed funding information from the acknowledgments section and have added a ‘Supporting Information’ section at the end of the manuscript. Supporting information documents as well as the manuscripts have been named accordingly. We have uploaded a manuscript with track changes (in red) and an identical manuscript without track changes. 

a) Did participants provide their written or verbal informed consent to participate in this study?

We have now addressed these comments, which were also shared by Reviewer 2. Please see our response to Reviewer 2 below regarding the text that we have added in our methods section in which we explain the procedure we followed to obtain participants’ consent. In short, consent was verbal after participants had read both the participant information sheet and consent form. Consent was obtained by participants stating it in the recording before the interviews commenced (hence it has been recorded). This form of consent was approved by the ethics committee due to COVID restrictions.

“The research presented here was supported by a QR seed grant by the School of Psychology and Life Sciences at Canterbury Christ Church University awarded to Evangelos Ntontis, and by a UKRI grant awarded to John Drury, Clifford Stott, Stephen Reicher, Fergus Neville and Evangelos Ntontis (ES/V005383/1).” Please state what role the funders took in the study. If the funders had no role, please state: "The funders had no role in study design, data collection and analysis, decision to publish, or preparation of the manuscript." If this statement is not correct you must amend it as needed. Please include this amended Role of Funder statement in your cover letter; we will change the online submission form on your behalf.

We have amended the financial disclosure, which now reads as: The research presented here was supported by a QR seed grant by the School of Psychology and Life Sciences at Canterbury Christ Church University awarded to Evangelos Ntontis, and by a UKRI grant awarded to John Drury, Clifford Stott, Stephen Reicher, Fergus Neville and Evangelos Ntontis (ES/V005383/1). The funders had no role in study design, data collection and analysis, decision to publish, or preparation of the manuscript.

“The research presented here was supported by a QR seed grant by the School of Psychology and Life Sciences at Canterbury Christ Church University awarded to Evangelos Ntontis, and by a UKRI grant awarded to John Drury, Clifford Stott, Stephen Reicher, Fergus Neville and Evangelos Ntontis (ES/V005383/1).”

“The research presented here was supported by a QR seed grant by the School of Psychology and Life Sciences at Canterbury Christ Church University awarded to Evangelos Ntontis, and by a UKRI grant awarded to John Drury, Clifford Stott, Stephen Reicher, Fergus Neville and Evangelos Ntontis (ES/V005383/1).”

We have now removed the financial statement from the main manuscript. The financial/funding statement that we would like to include is the following: The research presented here was supported by a QR seed grant by the School of Psychology and Life Sciences at Canterbury Christ Church University awarded to Evangelos Ntontis, and by a UKRI grant awarded to John Drury, Clifford Stott, Stephen Reicher, Fergus Neville and Evangelos Ntontis (ES/V005383/1). The funders had no role in study design, data collection and analysis, decision to publish, or preparation of the manuscript.

We have reviewed our reference list and verify that details are accurate, links are working, and no papers have been retracted since our paper was submitted. Thus, no changes to the reference list have been made. 

Comments from Reviewer 1

Reviewer #1: The study is quite interesting and also it is well-written article. I have thoroughly gone through the whole article and found it to be quite interesting paper with good policy implication. Authors have worked alot on all the sections of the paper and thus, I strongly support this article for publication without any further changes.

Thank you for your kind words and suggestion to accept the paper as it is. 

Comments from Reviewer 2

Reviewer #2: I would first like to extend appreciation for your work. Interesting set of thematic analyses. Please find below minor comments/questions:

1. Consent statements have not been included in your manuscript. While it is good that you've included the ethical statement, it would also be important to include the consent statement as well.

We have included the following statement in the method section: “Before the interviews commenced, participants had to provide the interviewers with their consent. Since the interviews were carried out online or while observing social distancing measures, a consent form was sent to participants in advance electronically together with a participant information sheet which described the aims of the project. Participants then had to confirm verbally that they read and understood the information sheet, that their participation was voluntary and that they could withdraw at any point without the need to provide a reason, that they agreed to the interview being audio recorded, that their information would be kept strictly confidential in line with the lead author’s University Research Privacy Notice, and that they agreed to take part in the project overall.”

2. Can the authors clearly state their exclusion criteria? (Before I read your "limitations" segment, I assumed that you deliberately focused on a specific demographic)

Thank you for this observation. Our inclusion criteria are discussed in the ‘participants, recruitment, and interview questions’ section, where we state that ‘Our only inclusion criteria were that participants were over 18 years old and resided in the United Kingdom at the onset and through the early course of the pandemic’. However, we also added the following sentences which make our sampling strategy more explicit: ‘We used opportunity sampling and participants were recruited through personal contacts, snowballing, and social media (Facebook, Twitter) on the basis of their willingness to participate in our study. Thus, the sample is not representative of the UK population’

3. I believe you must address the sample size as a limitation maybe, to note that it is in fact to small to be used for generalization of the topic under discussion.

We have added a few lines in our discussion where we highlight sample size limitations as well: “Third, our sample was relatively small (but nevertheless typical for semi-structured, qualitative interview studies), young, and not representative in terms of age, socioeconomic status, or geographical location. These factors might have affected participants’ risk perceptions and risk tolerance, affecting the findings. Based on the aforementioned limitations, future research should collect data at the time that stockpiling behaviours are observed and from larger and representative samples.”

---

## [Decision Letter · Decision Letter 1]

15 Feb 2022

Is it really “panic buying”? Public perceptions and experiences of extra buying at the onset of the COVID-19 pandemic

PONE-D-21-36415R1

Dear Dr. Ntontis,

We’re pleased to inform you that your manuscript has been judged scientifically suitable for publication and will be formally accepted for publication once it meets all outstanding technical requirements.

Kind regards,

Mumtaz Alam, PhD

Academic Editor

PLOS ONE

Additional Editor Comments (optional):

Reviewers' comments:

Reviewer's Responses to Questions

**Comments to the Author**

1. If the authors have adequately addressed your comments raised in a previous round of review and you feel that this manuscript is now acceptable for publication, you may indicate that here to bypass the “Comments to the Author” section, enter your conflict of interest statement in the “Confidential to Editor” section, and submit your "Accept" recommendation.

Reviewer #1: All comments have been addressed

Reviewer #2: All comments have been addressed

2. Is the manuscript technically sound, and do the data support the conclusions?

Reviewer #1: Yes

Reviewer #2: Yes

3. Has the statistical analysis been performed appropriately and rigorously? 

Reviewer #1: Yes

Reviewer #2: Yes

4. Have the authors made all data underlying the findings in their manuscript fully available?

Reviewer #1: Yes

Reviewer #2: Yes

5. Is the manuscript presented in an intelligible fashion and written in standard English?

Reviewer #1: Yes

Reviewer #2: Yes

6. Review Comments to the Author

Reviewer #1: After carefully reviewing the revision, i recommend this paper for final acceptance and appreciating their hardwork for producing good quality research.

Reviewer #2: I would like to extend my appreciation to the authors for taking the feedback into the betterment of the manuscript. All prior questions have been addressed. No further comments.

7. PLOS authors have the option to publish the peer review history of their article (what does this mean?). If published, this will include your full peer review and any attached files.

Reviewer #1: No

Reviewer #2: No

---

## [Editor Report · Acceptance letter]

17 Feb 2022

PONE-D-21-36415R1 

Is it really “panic buying”? Public perceptions and experiences of extra buying at the onset of the COVID-19 pandemic 

Dear Dr. Ntontis:

I'm pleased to inform you that your manuscript has been deemed suitable for publication in PLOS ONE. Congratulations! Your manuscript is now with our production department. 

Kind regards, 

on behalf of

Dr. Mumtaz Alam 

Academic Editor

PLOS ONE